# Validity and Applicability of Introducing a Healthcare-Associated Infection Surveillance System in Dental Hospitals in Korea Using the Delphi Technique

**DOI:** 10.3390/healthcare13233065

**Published:** 2025-11-26

**Authors:** Sun Young Jeong, So-Youn An

**Affiliations:** 1College of Nursing, Konyang University, Daejeon 35365, Republic of Korea; jsy7304@konyang.ac.kr; 2Department of Pediatric Dentistry, College of Dentistry, Wonkwang University, Iksan 54538, Republic of Korea

**Keywords:** dental hospital, healthcare-associated infection, surveillance, Delphi survey

## Abstract

**Background:** This study determined the validity and applicability of introducing a healthcare-associated infection surveillance system in dental hospitals using the Delphi technique. **Methods:** The Delphi questionnaire was developed by conducting a systematic literature review and focus group interview involving five dentists and dental hygienists with experience in performing infection control in dental hospitals and three infection control experts. The Delphi survey was administered to 16 experts, including 6 dentists and 5 dental hygienists with experience in infection control, 2 infection control nurses, and 3 infectious disease physicians. **Results:** The Delphi survey demonstrated a high level of agreement on the necessity of introducing a healthcare-associated infection surveillance system in dental hospitals and the need for dental hospitals to participate in the “Korean National Healthcare-associated Infections Surveillance System” that is currently operated on a national level. However, the level of agreement on the applicability was low given the lack of employees responsible for infection control. **Conclusions:** Surveillance criteria and a system for process indicators suitable for dental hospitals should be established. We suggest developing infection surveillance indicators and applying trials of the items with a high priority, as assessed in this study, to support dental hospitals’ participation in the “Korean National Healthcare-associated Infections Surveillance System”.

## 1. Introduction

Infection control in healthcare facilities is paramount to the safety of patients. It ensures the quality of medical services and includes protection of individuals exposed to the hospital environment, including patients, employees, and visitors [1]. Healthcare-associated infections (HAIs) were reported in 7–15% of patients admitted to acute care hospitals and have been associated with various economic losses, such as an increased length of hospital stay, high medical expenses, and medical disputes. The Korean National Healthcare-associated Infections Surveillance System (KONIS) calculates the incidence of HAIs in critical care units, neonatal intensive care units, and surgical sites with the voluntary participation of multiple medical centers. It also reports on performance rates of hand hygiene and prevention activities of central line-associated bloodstream infections. Since 2022, its operation has continued to expand, including the development of a long-term care hospital surveillance system [2].

Various surgical procedures, such as dental extractions and periodontal surgeries, are performed in dental hospitals. There is a high likelihood of infection in dental hospitals because oral bacteria or secretions, such as saliva and blood, are spread through air droplets. In addition, sharp instruments and various devices used in invasive procedures can be potential mediums of transmission. Because of the nature of dental settings, not only dental patients but also employees of dental hospitals are exposed to the risk of infection, necessitating systematic and sustainable infection control [3]. Moreover, contaminated items often come and go from the clinic to the dental laboratories and vice versa, and this increases the hazard, the possibility of microbial reservoirs, and the chance of infection transmission [4,5,6,7,8,9,10,11,12,13].

Previous studies [14,15,16] have investigated the knowledge and perception of infection control of employees of dental care centers and the current infection control procedures performed in dental medical centers to develop guidelines on infection control. However, little research has been conducted to establish HAI surveillance and a surveillance system to investigate the types and incidence of HAIs likely to occur in dental medical centers.

Therefore, this study aimed to investigate the validity and applicability of HAI surveillance systems in dental hospitals to gather basic data for establishing infection control policies and improvement initiatives in dental hospitals. Ultimately, the goal is to determine surveillance indicators that can continuously monitor the effects of infection control improvement initiatives and improve the quality of infection control in dental hospitals.

To this end, the Delphi method was used, and a panel of experts made a consensus decision.

## 2. Methods

### 2.1. Study Design

This study followed the Delphi method, which objectifies the intuitive judgment of experts when data on the specific topic are insufficient. The Delphi technique is a systematic and structured method that uses a series of questionnaires to extract and synthesize comments and judgments of experts and obtain their final consensus opinion [17]. Therefore, herein, a Delphi survey was administered to assess the validity and applicability of introducing an HAI surveillance system in dental hospitals by collecting the opinions of an expert group.

### 2.2. Study Population

This study included 17 dentists or dental hygienists who had at least 5 years of experience in performing infection control-associated duties and were recommended by the Korean Dental Hospital Association (KDHA) or the Korea Dental Infection Control Association (KDICA). Five infection control doctors or nurses who had at least 5 years of experience in performing infection surveillance in the infection control unit of healthcare centers and were recommended by the Korean Society of Health-associated Infection Control (KOSHIC) were also included. Consequently, the Delphi survey was administered to a total of 22 participants.

### 2.3. Preparation of the Delphi Survey Questions

A systematic literature review and focus group interview (FGI) were conducted to develop questions for a preliminary instrument to assess the validity and applicability of the HAI surveillance system in dental hospitals. PubMed (https://pubmed.ncbi.nlm.nih.gov/), EMBASE (https://www.embase.com/), Cochrane Library (https://www.cochranelibrary.com/), and RISS (https://www.riss.kr/) were used for the literature search. Core questions included the following: (1) What are the types, incidences, and characteristics of HAIs reported in dental hospitals in Korea and internationally? (2) What are the types and characteristics of HAI surveillance systems used in dental hospitals in Korea and internationally? Keywords related to the core research questions included dentistry and dental hospitals, HAIs, and HAI surveillance systems. For keywords related to dentistry and dental hospitals, MeSH and Emtree terms for the following were included: “Dental Service, Hospital,” “Dental Clinics,” “Dental Care,” “Oral Surgical Procedures,” “Tooth Extraction,” “Surgery, Oral,” “Maxillofacial Prosthesis Implantation,” “Maxillofacial Prosthesis,” “Dental Implants,” “Dental Implantation,” “Dental Scaling,” “Dentistry,” “Dentists,” “Dental Auxiliaries,” and “Dental Hygienists.” For HAIs, MeSH and Emtree terms for “Cross Infection” were selected as search terms. For HAI surveillance systems, MeSH and Emtree terms for “Safety Management” were selected as the search terms.

To participate in the FGI, individuals were required to have work experience in responding to infectious diseases in healthcare centers. Based on a previous study [18], a group of 6–12 participants with common characteristics was required for an appropriate discussion. Accordingly, a total of eight participants, including five members of the KDHA or KDICA with at least 5 years of experience in infection control-related work or consultation in dental hospitals and three infection control doctors or nurses with at least 5 years of experience in operating the KONIS, were included in the FGI.

The FGI consisted of questions to determine the types and current status of HAIs that have occurred or are likely to occur in dental hospitals and the validity and applicability of the HAI surveillance system. Items included the types and incidences of HAIs commonly reported in dental hospitals; HAI surveillance systems operated in dental hospitals, whether they are monitoring HAIs or the degree of performance of infection control (for example, the performance of hand hygiene) as a part of the infection control system; if so, the types and methods of surveillance; the likelihood of onset of HAIs; impact on patients and employees; what HAI surveillances are considered necessary in consideration of their preventive potential; is it necessary to be involved in the KONIS; and if so, what are the obstacles to participating in the KONIS and ways of overcoming it.

Based on the results of the systematic literature review and FGI, we developed the Delphi questionnaires regarding the validity and applicability of the HAI surveillance system. The survey consisted of 31 items determining the level of agreement on a 9-point scale, ranging from 1 (strongly disagree) to 9 (strongly agree), with 5 points being neutral. These included 15 items regarding characteristics of HAIs in dental hospitals, 6 concerning HAI surveillance, 2 about the safety of employees, 8 pertaining to the surveillance of infection control processes, and 2 regarding HAI surveillance systems. In case of disagreement among panel experts, they were requested to explain the reasons in written form. Also, they were asked to write an improvement proposal, if appropriate, even in cases of agreement. The survey contained two descriptive questions, and responders were asked to describe the difficulties with participating in the KONIS and the requirements for participation.

### 2.4. Delphi Survey

The Delphi survey was administered two times between December 2023 and February 2024. We emailed the survey and asked the receivers to complete and return it by the deadline. The Delphi panel included a total of 22 experts, including 17 dentists or dental hygienists and 5 infection control nurses.

The first round of the survey was administered from 9 December 2023 to 9 January 2024. From 1 to 15 February 2024, the mean, median, and content validity ratio (CVR) values and the comments on their responses from the first survey were emailed to the respondents. In the second round of the survey, they were provided with an opportunity to modify the degree of their agreement and comment on the validity and applicability of the HAI surveillance system by checking the results from their first responses.

### 2.5. Data Analysis

Data collected from the Delphi survey were analyzed using the equation proposed by Lawshe [19]. The number of experts who participated in the second Delphi round was 16, and the minimum CVR was ≥0.49. Items with a minimum CVR value of <0.49 were considered to have low validity [19].

### 2.6. Ethical Considerations

This study was approved by the Konyang University Hospital Institutional Review Board (KYU 2023-05-048-001). In the first email sent to potential participants, they were asked to provide consent to participate in the Delphi survey. During the process, we ensured that participants could freely decide to participate in the study and guaranteed their anonymity. In addition, participants were informed of their right to refuse or withdraw from participation in the study at any time; their data would be immediately disposed of in such an event. The identity of the experts who participated in the survey was not disclosed to other experts during the data collection process. The survey distribution emails were sent to each expert individually.

## 3. Results

### 3.1. Characteristics of the Panel of Experts Responding to the Survey

Among the 22 experts included, only 16 responded (response rate: 72.7%). The mean age of the 16 experts participating in the Delphi survey was 47.88 ± 6.75 years, with 18.8% identified as male and 81.3% identified as female. Of the responders, 25.0%, 43.8%, 18.8%, and 12.5% were dentists, dental hygienists, infectious disease physicians, and infection control nurses, respectively.

### 3.2. First-Round Delphi Survey Results

According to the Delphi survey responses of 16 experts, the validity and applicability of introducing HAI surveillance systems in dental hospitals are shown in Table 1, Table 2, Table 3, Table 4 and Table 5. The 15 items showed strong agreement, with a mean value of 7 points, median value of 7 points, and CVR score of 0.49.

#### 3.2.1. Characteristics of HAIs in Dental Hospitals

For item 3 [preventability of postoperative (postprocedural) infection at the surgical (procedural) site in dental hospitals], the mean, median, and CVR values were 7.25, 7, and 0.63 points, respectively. Also, a comment was made that infections are preventable via postoperative precautions and reservation management. For item 9 (preventability of infections associated with dental unit waterline), the mean, median, and CVR values were 7.44, 8, and 0.75 points, respectively. Further, a suggestion was made that, since pathogens can exist on dental line waterlines, water quality should be controlled to prevent surgical site infections and antibiotic resistance. For item 15 (prevention of exposure to respiratory secretions), the mean, median, and CVR values were 7.19, 7, and 0.88 points, respectively. A suggestion was made that HAIs can be prevented by identifying related symptoms and complying with infection control measures [hand hygiene, personal protective equipment (PPE), disinfecting surfaces, ventilation, and respiratory etiquette] when healthcare providers examine patients. The three items had interquartile ranges of ≤1.25, indicating consistent expert opinions and a moderate or higher level of consensus.

However, for items related to dentistry (item numbers 1, 2, 4–8, and 11–14), the degree of agreement did not reach the standard point (mean and median ≥7 points, respectively, CVR ≥ 0.49) (Table 1).

**Table 1 healthcare-13-03065-t001:** Delphi survey results on infection surveillance in dental hospitals: Healthcare associated infection characteristics.

Item Number	Delphi Items	First Delphi Survey (n = 16)				Second Delphi Survey (n = 16)			
		Mean	Median	CVR	IQR	Mean	Median	CVR	IQR
1	Postoperative (postprocedural) surgical (procedural) site infection often occurs in a dental hospital.	3.00	2.00	−0.63	1.75	2.60	2.00	−0.88	1.25
2	Postoperative (postprocedural) surgical (procedural) site infection severely affects patients in a dental hospital.	4.94	5.00	−0.50	3.25	5.10	5.00	−0.50	2.00
3 *	Postoperative (post-procedural) infection at the surgical (procedure) site is preventable in a dental hospital.	7.25	7.00	0.63	1.00	7.30	7.00	−0.63	1.00
4	Postoperative (postprocedural) antibiotic-resistant infection often occurs in a dental hospital.	2.56	2.00	−0.88	1.25	2.70	2.00	−0.88	1.00
5	Postoperative (postprocedural) antibiotic-resistant infection severely affects patients.	5.44	5.50	−0.25	3.50	5.30	5.00	−0.63	2.00
6	Postoperative (postprocedural) antibiotic-resistant infection is preventable in dentistry.	6.25	7.00	0.13	3.00	6.30	6.50	0.00	2.00
7	Infections associated with dental unit waterlines often occur in a dental hospital.	2.69	2.00	−0.88	2.25	2.50	2.00	−0.88	1.00
8	Infections associated with dental unit waterline severely affect patients.	5.63	6.50	0.00	3.00	5.80	6.50	0.00	2.25
9 *†	Infections associated with dental unit waterlines are preventable.	7.44	8.00	0.75	1.25	8.00	8.00	1.00	0.25
10 †	Employees of dentistry often experience sharp instrument injuries.	6.94	7.50	0.50	1.75	7.30	8.00	0.50	1.25
11	Sharp instrument injuries severely affect employees.	5.31	5.50	−0.13	4.00	5.40	6.00	−0.38	4.00
12	Sharp instrument injuries are preventable.	7.50	8.00	0.38	2.00	7.30	7.50	0.38	2.00
13	Employees of dental hospitals are often in contact with patients with respiratory symptoms.	6.44	6.00	−0.13	2.25	6.90	7.00	0.13	1.00
14	Exposure of an employee to a patient with respiratory symptoms severely affects the employee.	6.44	6.00	−0.13	3.00	6.60	6.00	−0.13	1.25
15 *	An accident exposed to respiratory secretions is preventable in dental hospitals.	7.19	7.00	0.88	1.00	6.90	7.00	0.75	1.00

CVR, Content validity ratio. IQR, Interquartile Range. * Items with the mean of ≥7 points, median of ≥7 points, and CVR of ≥0.49 in the first Delphi survey. † Items with the mean of ≥7 points, median of ≥7 points, and CVR of ≥0.49 in the second Delphi survey.

#### 3.2.2. HAI Surveillance

For item 20 (the necessity for surveillance of infections associated with dental unit waterlines), the mean, median, and CVR values were 7.44, 8, and 0.5 points, respectively. A suggestion was made that infections associated with waterlines that have direct contact with the mouth of patients should be monitored. For item 21 (applicability of surveillance on dental unit-associated infections), the mean, median, and CVR values were 7.13, 8, and 0.5 points, respectively. Suggestions were made that waterline surveillance should be undertaken by culturing the surface of waterlines, and diagnostic criteria for dental unit waterline-associated infections should be established. The interquartile range for the two items were 2.5, indicating a relatively wide variability among expert responses.

However, the degree of agreement did not reach the standard point for items such as the applicability of surveilling surgical (procedural) site infections and the necessity and applicability of surveilling infections due to antibiotic resistance in dental hospitals (Table 2).

**Table 2 healthcare-13-03065-t002:** Delphi survey results on infection surveillance in dental hospitals: Healthcare associated infection surveillance.

Item Number	Delphi Items	First Delphi Survey (n = 16)				Second Delphi Survey (n = 16)			
		Mean	Median	CVR	IQR	Mean	Median	CVR	IQR
16 †	Surveillance of postoperative (postprocedural) infection on the surgical (procedural) site ‡ is necessary for patient safety in dentistry.	7.38	7.50	0.38	2.25	7.80	8.00	0.75	1.00
17	Surveillance of surgical (procedural) site infection ‡ is applicable to dental hospitals.	5.94	7.00	0.13	3.00	6.10	7.00	0.13	2.00
18	Surveillance of postoperative (postprocedural) antibiotic-resistant infections ‡ is required for patient safety in dental hospitals.	5.94	6.00	−0.25	2.00	6.20	6.00	−0.13	2.00
19	Surveillance of postoperative (postprocedural) antibiotic-resistant infection ‡ is applicable to dental hospitals.	5.13	5.00	−0.50	2.50	5.10	5.00	−0.63	1.00
20 *†	Surveillance of dental unit waterline-associated infections ‡ is required for patient safety in dental hospitals.	7.44	8.00	0.50	2.50	7.40	8.00	0.50	1.25
21 *†	Surveillance of dental unit waterline-associated infections ‡ is applicable to dental hospitals.	7.13	8.00	0.50	2.50	7.40	8.00	0.50	2.50

CVR, Content validity ratio. IQR, Interquartile Range. * Items with the mean of ≥7 points, median of ≥7 points, and CVR of ≥0.49 in the first Delphi survey. † Items with the mean of ≥7 points, median of ≥7 points, and CVR of ≥0.49 in the second Delphi survey. ‡ Infection surveillance: investigating the onset of infection and its incidence and giving an individual, department, member, and executive feedback.

#### 3.2.3. Safety of Employees

For item 22 (the necessity of surveillance on sharp instrument injuries), the level of agreement was strong, with mean, median, and CVR values of 7.94, 8, and 0.75 points, respectively. For item 23 (sharp injury surveillance is applicable), the mean, median, and CVR values were 7.38, 7.5, and 0.63 points, respectively. The interquartile ranges (IQR) were 1.25 and 2.00 for items 22 and 23, indicating higher consensus for item 22 and greater variability for item 23. However, both items demonstrated high consensus with an IQR of 1.00 in the second Delphi round (Table 3).

**Table 3 healthcare-13-03065-t003:** Delphi survey results on infection surveillance in dental hospitals: Safety of employees.

Item Number	Delphi Items	First Delphi Survey (n = 16)				Second Delphi Survey (n = 16)			
		Mean	Median	CVR	IQR	Mean	Median	CVR	IQR
22 *†	Surveillance of sharp instrument injuries is required for the safety of employees in dental hospitals.	7.94	8.00	0.75	1.25	8.30	8.00	1.00	1.00
23 *†	Surveillance of sharp instrument injuries is applicable to dental hospitals.	7.38	7.50	0.63	2.00	7.60	7.50	0.88	1.00

CVR, Content validity ratio. IQR, Interquartile Range. * Items with the mean of ≥7 points, median of ≥7 points, and CVR of ≥0.49 in the first Delphi survey. † Items with the mean of ≥7 points, median of ≥7 points, and CVR of ≥0.49 in the second Delphi survey.

#### 3.2.4. Surveillance of Infection Control Processes

For item 24 (the necessity of hand hygiene surveillance), the mean, median, and CVR values were 8.13, 8.5, and 0.88 points, respectively. For item 25 (applicability of hand hygiene surveillance), the mean, median, and CVR values were 7.69, 8, and 0.5 points, respectively. Items 24 and 25 showed a high level of agreement. For item 26 (the necessity of surveillance on waterlines), the mean, median, and CVR values were 8.39, 9.0, and 1.0 points, respectively. A suggestion was made that disinfection of waterlines and monitoring of water quality are required for patient safety since water use is not optional while providing dental care.

For item 28 (surveillance on cleaning/disinfection/sterilization of dental instruments is necessary for patient safety), the mean, median, and CVR values were 8.31, 9.0, and 0.88 points, respectively. A suggestion was made that dental instruments must undergo the decontamination processes (cleaning/disinfection/sterilization) because cross-contamination can occur owing to the instruments being exposed to saliva or blood. Another suggestion was to monitor whether the process was accurate. For item 29 (surveillance on cleaning/disinfection/sterilization of dental instruments is applicable), the mean, median, and CVR values were 7.69, 7.5, and 0.63 points, respectively.

For item 30 [surveillance on environmental surface disinfection (control) is necessary for patient safety], the mean, median, and CVR values were 8.0, 8.0, and 0.88 points, respectively. A suggestion was made that surface disinfection is an important measure because interest in infection control has increased since the coronavirus disease (COVID-19) pandemic, and patients are unaware of whether they are infected or do not inform healthcare providers of their infection. For item 31 (surveillance on environmental surface disinfection is applicable), the mean, median, and CVR values were 7.5, 7.5, and 0.5 points, respectively. The interquartile ranges of the eight items ranged from 1 to 2.25, but all decreased to ≤1.00 in the second Delphi round, indicating a high level of consensus (Table 4).

**Table 4 healthcare-13-03065-t004:** Delphi survey results on infection surveillance in dental hospitals: Surveillance of infection control processes.

Item Number	Delphi Items	First Delphi Survey (n = 16)				Second Delphi Survey (n = 16)			
		Mean	Median	CVR	IQR	Mean	Median	CVR	IQR
24 *†	Hand hygiene surveillance (monitoring and feedback) is necessary for patient safety in dental hospitals.	8.13	8.50	0.88	1.25	8.40	9.00	1.00	1.00
25 *†	Hand hygiene surveillance (monitoring and feedback) is applicable to dental hospitals.	7.69	8.00	0.50	2.25	8.10	8.00	0.88	1.00
26 *†	Surveillance on disinfection of waterlines (monitoring and feedback) is necessary for patient safety in dental hospitals.	8.38	9.00	1.00	1.00	8.70	9.00	1.00	1.00
27 *†	Surveillance on disinfection (monitoring) of waterlines is applicable to dental hospital.	8.25	9.00	0.88	1.00	8.60	9.00	1.00	1.00
28 *†	Surveillance (monitoring and feedback) on cleaning/disinfection/sterilization of dental instruments is necessary for patient safety in dental hospitals.	8.31	9.00	0.88	1.00	8.80	9.00	1.00	0.25
29 *†	Surveillance (monitoring and feedback) on cleaning/disinfection/sterilization of dental instruments is applicable to dental hospitals.	7.69	7.50	0.63	2.00	7.70	7.50	0.88	1.00
30 *†	Surveillance (monitoring and feedback) on environmental surface disinfection (control) is necessary for patient safety in dental hospital.	8.00	8.00	0.88	2.00	8.40	8.50	1.00	1.00
31 *†	Surveillance (monitoring and feedback) on environmental surface disinfection (control) is applicable to dental hospitals.	7.50	7.00	0.50	2.25	7.50	7.00	0.88	1.00

CVR, Content validity ratio. IQR, Interquartile Range. * Items with the mean of ≥7 points, median of ≥7 points, and CVR of ≥0.49 in the first Delphi survey. † Items with the mean of ≥7 points, median of ≥7 points, and CVR of ≥0.49 in the second Delphi survey.

#### 3.2.5. Perspectives on Participation in KONIS: Necessity, Challenges, and Requirements

For items 32 and 33, which assessed the necessity of dental hospital participation in KONIS, the first-round Delphi results did not meet the consensus criteria (Table 5).

For item 34 (what are difficulties for dental hospitals to be involved in the “KONIS”), the panel complained about the following: (1) inadequate infection surveillance system in dental hospitals; (2) the lack of awareness for the risk of infection; (3) an increased number of outpatients due to the nature of dental hospitals; (4) inapplicability of acute care hospital-oriented surveillance systems: a consensus on diagnostic criteria for HAIs is necessary since the most common pathogen in HAIs is resident bacteria within the mouth; (5) a lack of manpower and resources of infection surveillance; and (6) absence of compensation system for infection control.

For item 35 (what does a dental hospital need to be involved in the “KONIS”), the panel suggested the following: (1) manpower and training on infection control: secure professional manpower that is responsible for infection control in dental hospitals and develop education and training programs for them; (2) establish surveillance system and indicators: establish surveillance system and indicators for infection control that apply to dental hospitals, along with methods to standardize and manage the same; (3) need for establishing legal and governmental support and compensation system; (4) propose a plan for infection control given the nature of dental hospitals; (5) reinforce education and improve awareness of infection control; and (6) propose the appropriation of the fee for infection prevention and introduction of incentives.

**Table 5 healthcare-13-03065-t005:** Delphi survey results on the necessity of introducing KONIS in dental hospitals.

Item Number	Delphi Items	First Delphi Survey (n = 16)				Second Delphi Survey (n = 16)			
		Mean	Median	CVR	IQR	Mean	Median	CVR	IQR
32 †	It is necessary to introduce the KONIS to dental hospitals.	7.31	7.00	0.38	3.00	7.60	7.00	0.63	2.00
33 †	A dental hospital is required to be involved in the “KONIS.”	6.94	7.00	0.25	2.25	7.40	7.00	0.50	2.25

CVR, Content validity ratio. IQR, Interquartile Range. KONIS, The Korean National Healthcare-associated Infections Surveillance System. † Items with the mean of ≥7 points, median of ≥7 points, and CVR of ≥0.49 in the second Delphi survey.

### 3.3. Second-Round Delphi Survey Results

The second round of the Delphi survey was administered to 16 panel experts. The mean, median, and CVR values for the 17 items were ≥7.0, ≥7.0, and ≥0.49, respectively (15 items in the first round of the Delphi survey). Compared to item 3 (preventability of postoperative infections at the surgical site) and item 15 (preventability of accidental exposure to respiratory secretions) from the first round Delphi survey, which had mean, median, and CVR values higher than the standard values (≥7, ≥7, ≥0.49 points, respectively), the corresponding values from the second round Delphi survey were lower than the standard value, resulting in a lower level of agreement between the experts (Table 1).

In the second round of the Delphi survey, four additional items had mean, median, and CVR values higher than the standard values. These were item 10 (incidence of sharp instrument injuries), item 16 (surveillance on postoperative infection on the surgical site is necessary for patient safety), item 32 (it is necessary to introduce the KONIS to dental hospitals), and item 33 (dental hospitals are required to be involved in the “KONIS”).

For item 10, the mean, median, and CVR values were 7.3, 8.0, and 0.5 points, respectively (Table 1). A suggestion was made that although the incidence of secondary infections caused by sharp injuries among dental employees is rare, the frequency of sharp instrument injury occurrence is high. For item 16, the mean, median, and CVR values were 7.8, 8.0, and 0.75 points, respectively (Table 2). It was suggested that it is important to establish the diagnostic criteria for surgical (procedural) site infection in advance, investigate the onset of infection and its incidence, and understand the current status. Nonetheless, it is difficult to conduct immediate surveillance on surgical (procedural) site infection owing to the lack of manpower.

For item 32 (it is necessary to introduce the KONIS to dental hospitals), the mean, median, and CVR values were 7.6, 7.0, and 0.63 points, respectively. The following comments were made: the oral cavity and the skin surrounding it can be the key route of infection because surgeries or procedures are performed in dentistry in this area; employees are always exposed to infections; and sharp instrument injuries occur frequently. Accordingly, the introduction of an HAI surveillance system to dentistry is essential. For item 33 (dentistry in Korea is required to be involved in the “KONIS”), the mean, median, and CVR values were 7.4, 7.0, and 0.5 points, respectively (Table 5). A suggestion was made that since dentistry has a different operating system from an acute care medical center, an independent infection surveillance system is required rather than participating in the existing KONIS.

## 4. Discussion

As life expectancy increases, the prevalence of Alzheimer’s disease will increase even further. Dentistry seems to be in the first line of prevention and should begin to equip itself with skills and updated knowledge for taking care of the different needs, demands, and aspirations of typically aged and Alzheimer’s patients, including innovation through digital dentistry [20]. But the guidelines for dentistry published by the CDC since 2003 include, as a Special Consideration, a subsection called Dental Laboratory, but it has not been updated in later versions [21,22]. In this study, we developed Delphi survey items to evaluate the validity and applicability of the introduction of HAI surveillance systems to dentistry through a systematic literature review and FGI. According to the results of the systematic literature review and FGI, HAIs that could occur in dental hospitals include (1) surgical site infection, (2) antibiotic-resistant infections, (3) infections associated with contaminated dental unit waterlines, (4) infections caused by sharp instrument injuries in employees, and (5) respiratory infections in employees.

For postoperative surgical site infection, the mean, median, and CVR values of all items pertaining to the probability of onset, effects on patients, and preventability did not reach the standard values (mean of ≥7 points, median of ≥7 points, and CVR of ≥0.49 points). Previous studies have reported hepatitis C transmission in patients who underwent dental surgeries [23], the onset of *Mycobacterium* abscess after undergoing treatment of dental roots [24], and the onset of surgical wound infections in patients following oral and mandibular/maxillary surgeries [25]. Since the participants have limited experience with surgical site infections, this study might not accurately reflect the scale of infections because the incidence or prevalence of the infection has not been investigated. Dental surgeries and procedures have a higher risk of surgical site infections due to the resident bacteria within the mouth; thus, it is necessary to investigate cases of actual infection following surgery via infection surveillance.

Regarding antibiotic-resistant infections, the levels of agreement for all items relating to the probability of onset, effects on patients, and preventability did not reach the standard values. Short-term use of antibiotics is common in dentistry; however, long-term use of antibiotics is rare. Since most surgeries are performed in the mouth, and resident bacteria within the mouth are the most common causative bacteria, the participants concluded that there is low concern about antibiotic-resistant infections. However, despite the infrequency of surgical site infections, postoperative infections lead to an increase in the length of hospital stay and medical expenses. Moreover, antibiotic-resistant infections can severely affect patients. They can be prevented by fully explaining the dosage regimen of antibiotics and prescribing only the required dose [26]. Regarding infections associated with contaminated dental unit waterlines, the level of agreement on the possibility of onset and effects on patients did not reach the standard values; nonetheless, the level of agreement was high for the item relating to preventability. Although previous studies reported pneumonia associated with a dental unit waterline [27] and dentistry-associated *Legionella* infection [28], none of the cases was a cluster infection. The decreased level of agreement on the probability of onset and effects of infections associated with contaminated dental unit waterlines on patients is because it is difficult to epidemiologically prove whether the patients were infected from the contaminated dental unit waterlines. However, the level of agreement on preventability was high in this study. This is likely because contaminated water used in dental procedures is more likely to cause infection in immunocompromised patients or patients undergoing various invasive dental procedures, emphasizing the need for prevention by disinfecting waterlines and performing microbiological analysis of dental unit waterlines [29].

The level of agreement regarding the probability of onset of infections associated with sharp instrument injuries in employees was high; however, the level of agreement on its effects on employees and preventability did not reach the standard values. Transmissions of blood-borne pathogens in a dental healthcare setting [30] have rarely been reported, and none caused cluster infections. However, in a dental setting, sharp injuries are more likely to occur because most of the instruments are sharp and must be manipulated in the narrow environment of the mouth. Despite the concern over infections owing to sharp injuries, dental employees have limited awareness of appropriate measures and future management strategies [31]. Since prevention and management strategies for sharp instrument injuries are limited due to a large number of patients and a lack of manpower, education and training programs about their prevention for dental employees are required.

Regarding respiratory infections in the second round Delphi survey, the level of agreement on all items about the probability of onset, effects on employees, and preventability did not reach the standard values. With regard to the reported HAIs in dentistry healthcare providers, a total of 443 were infected with COVID-19 in 2020 and 2021, of whom 129 were dentists, 291 were dental hygienists, and 23 were dental technicians [32]. Having contact with patients with symptoms of respiratory infections without mask-wearing can affect healthcare providers in dentistry. However, it is challenging to determine cases of actual transmission of infection to healthcare providers after the treatment. Participants in this study suggested that practicing basic infection control, such as vaccination, wearing a mask, and ensuring proper ventilation, can protect employees from serious infections.

Regarding the necessity of surveillance on HAI in dental hospitals, the CVR value of surveillance on surgical site infections was higher in the second round than in the first-round Delphi survey, demonstrating a higher level of agreement. However, the level of agreement on applicability was low. Although there was agreement on the necessity for infection prevention that can be established and implemented through infection surveillance, the participants concluded that it is difficult to apply owing to inadequate diagnostic criteria for surgical site infection, absence of manpower for appropriate surveillance, and inadequate training and quality control for surveillance. Moreover, since most patients were outpatients, it could be hard to follow up with the patients to determine whether they were infected during a certain period after surgery.

Despite the low level of agreement on the applicability of HAI surveillance in this study, the level of agreement on the necessity and applicability of surveillance of process indicators, including hand hygiene, waterline disinfection, cleaning/disinfection/sterilization of dental instruments, and environmental surface disinfection, was high. Experts agreed with these items, and their importance emerged in terms of patient safety. Several previous studies investigated hand hygiene performance [33]; the vaccination rate for hepatitis B; the rate of wearing medical gloves, masks, and goggles [34]; the level of donning PPE while cleaning instruments; the performance rate of cleaning and disinfecting dental instruments; and the level of dental unit surface and waterline management [35]. However, it was difficult to identify cases where dental hospitals implement national surveillance systems for process indicators.

Regarding the necessity of introducing an HAI surveillance system in dental hospitals in Korea and whether dentistry should be a participant of “KONIS,” the level of agreement was higher in the second round compared to that in the first round of the Delphi survey. The nature of the Delphi survey is that the results of the first round are compared to one’s own opinion, and then the second round is completed [17]. In other words, the results of the first round positively affect the results of the second round. Since there is a high likelihood of being infected following exposure to saliva and blood and the use of invasive procedures in dental care settings, introducing an HAI surveillance system with regard to invasive surgery is required. Given the nature of dental hospitals, a practical surveillance system appropriate for dental surgery should be established, and dental hospitals can reinforce the national infection surveillance system by being involved.

The findings of this study demonstrate the need to introduce a healthcare-associated infection (HAI) surveillance system in dental hospitals. International surveillance systems—including the U.S. CDC’s National Healthcare Safety Network (NHSN) for surgical site infection surveillance [36], the WHO core components for infection prevention and control (IPC) [37], and the European Center for Disease Prevention and Control (ECDC) HAI-Net—commonly emphasize standardized case definitions, monitoring of process indicators, and national surveillance strategies that encompass outpatient and ambulatory care settings [38]. These approaches suggest that, to manage key risk factors unique to dental care environments—such as waterline quality, instrument sterilization, and occupational exposures—a surveillance model tailored specifically to dental settings is necessary.

Difficulties for dental hospitals participating in the KONIS included a lack of awareness among dental care providers about infection surveillance and infection control, a lack of manpower for surveillance, inadequate surveillance training, and an absence of surveillance criteria and programs. Accordingly, for a dental hospital to participate in the KONIS, the following plan is required: strengthening of the infection control program for employees of dentistry and improvement of awareness, training and education on infection control, experience sharing, cooperation between employees responsible for infection control in dentistry, selection of surveillance indicators and establishment of a surveillance criteria, creation of surveillance system, and political and financial support for infection control in dental hospitals.

From a policy perspective, effective implementation of an HAI surveillance system in dental hospitals requires several key actions: establishing dental-specific HAI definitions and a KONIS dental module, providing national IPC training for dental personnel, and introducing financial incentives to encourage participation. These measures align with the WHO IPC core components—governance, workforce capacity, and sustainable resources—and will enable dental hospitals to contribute more effectively to national HAI surveillance and strengthen patient safety.

## 5. Conclusions

Infections associated with dental surgeries and procedures are directly linked to patient safety. HAI surveillance systems, currently operated on a national level, require the participation of dental hospitals, and the results of our Delphi surveys showed a high level of agreement on the participation of dental hospitals. However, given the nature of dental hospitals, in which treatment is conducted on a unit chair rather than a bed, and the lack of employees responsible for infection control, a surveillance system appropriate for use in dental care settings should be established, and employee education and training for infection surveillance, surveillance criteria, and infection surveillance systems should be created.

Surveillance of process indicators, such as hand hygiene, waterline disinfection, cleaning/disinfection/sterilization of dental instruments, and environmental surface disinfection, is essential for patient safety, and the level of agreement on applicability in Korea was high. Since the medical care environment in dental hospitals differs from that in acute care medical centers, surveillance criteria and systems for process indicators suitable for dental hospitals should be established. The surveillance system on a national level can be reinforced by the participation of dental hospitals. To support dental hospitals in participation, we suggest developing infection surveillance indicators and applying trials of the items with a high priority, as assessed in this study.

## Data Availability

The original contributions presented in this study are included in the article. Further inquiries can be directed to the corresponding author.

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
