# Peer review of "Validity and Applicability of Introducing a Healthcare-Associated Infection Surveillance System in Dental Hospitals in Korea Using the Delphi Technique"

_healthcare, 2025, doi:10.3390/healthcare13233065_

Round 1
Reviewer 1 Report
Comments and Suggestions for Authors
The work aimed to investigate the feasibility of applying a national healthcare-associated infection (HAI) surveillance system (KONIS) to dental hospitals in Korea using the Delphi method. It involved conducting a two-round Delphi survey including 22 experts (dentists, hygienists, infection control specialists). Their main results indicated a strong agreement on the significant need of introducing such healthcare-associated infection (HAI) surveillance system in dental hospitals. Furthermore, insufficient manpower and lack of infection control infrastructure presented a moderate applicability agreement. However, high consensus on the importance of process surveillance indicators (hand hygiene, instrument sterilization, surface and waterline disinfection). The study concluded ion the need for developing tailored surveillance indicators and piloting them in dental hospitals to enable eventual participation in the national KONIS program.
The work is overall good and is of relevance to the readership of the journal. However, I have to make the following comments:
- The discussion could benefit from stronger linkage to global HAI surveillance literature and more practical policy implications.
- English usage and figure/table presentation require some editing to enhance their clarity.
- Inclusion of consensus stability (e.g., interquartile range, Kendall’s W) could strengthen the robustness of the findings.
- Table 1 is lengthy and consider splitting it or summarizing items for clarity.
Author Response
|
Thank you very much for taking the time to review this manuscript., “Validity and Applicability of Introducing a Healthcare-Associated Infection Surveillance System in Dental Hospitals in Korea Using the Delphi Technique”. We have revised the manuscript according to the Reviewer’s suggestions. We acknowledge that the quality of our manuscript was improved by the scrutinizing efforts of the reviewers and editors. The changes within the revised manuscript were highlighted (in red).
Point-by-point responses to the reviewers’ comments are provided below. Comments 1: The discussion could benefit from stronger linkage to global HAI surveillance literature and more practical policy implications. Author Response : Thank you for your thoughtful suggestions. In response, we have revised the discussion section by incorporating references to the surgical site infection surveillance systems of the U.S. CDC and the ECDC, as well as the World Health Organization’s emphasis on surveillance within its core components of infection prevention and control. In addition, policy recommendations have been added to the final paragraph to strengthen the practical implications of our findings.
Comments 2: English usage and figure/table presentation require some editing to enhance their clarity.
Comments 3: Inclusion of consensus stability (e.g., interquartile range, Kendall’s W) could strengthen the robustness of the findings. Author Response : Thank you for pointing this out. We included the interquartile range values, which indicate the stability of consensus, in the tables and described them in the results section.
Comments 4: Table 1 is lengthy and consider splitting it or summarizing items for clarity.
|
- Additional Notes
We sincerely hope that the revisions and the additional information we have provided sufficiently address the reviewer’s concerns. We are grateful for the opportunity to improve our manuscript.
Reviewer 2 Report
Comments and Suggestions for Authors
The reviewed work concerns the subject of Validity and Applicability of Introducing a Healthcare-Associated Infection Surveillance System in Dental Hospitals in Korea Using the Delphi Technique.
The initiative and idea are undoubtedly very good.
My biggest complaint is that the authors also omitted an important medical group: dental technicians. This group was only mentioned in line 342.
The field of dental prosthetics is very important because life expectancy has increased and a large group of patients use prosthetic restorations. In the case of prosthetic work, dental impressions pose a very serious threat as they are a breeding ground for bacteria. They often contain blood on their surface, which poses another risk of infection. Therefore, it would be necessary to supplement the publication with information related to this field. The authors should expand their work to include issues related to dental prosthetics. In addition to sterilizing tools, the process of taking an impression should be included. The issue of denture repair is also important, as dental technicians are at risk of infections related to the handling of dentures or other prosthetic restorations.
Additionally, the introduction should be more extensive and include more literature. Currently, it only has 7 items of literature.
Author Response
|
1. Summary |
|
|
|
Thank you very much for taking the time to review this manuscript., “Validity and Applicability of Introducing a Healthcare-Associated Infection Surveillance System in Dental Hospitals in Korea Using the Delphi Technique”. We have revised the manuscript according to the Reviewer’s suggestions. We acknowledge that the quality of our manuscript was improved by the scrutinizing efforts of the reviewers and editors. The changes within the revised manuscript were highlighted (in red).
Point-by-point responses to the reviewers’ comments are provided below. |
||
|
|
|
|
|
2. Point-by-point response to Comments and Suggestions for Authors
Reviewer 2 : The reviewed work concerns the subject of Validity and Applicability of Introducing a Healthcare-Associated Infection Surveillance System in Dental Hospitals in Korea Using the Delphi Technique.
The initiative and idea are undoubtedly very good.
|
||
|
Comments 1: My biggest complaint is that the authors also omitted an important medical group: dental technicians. This group was only mentioned in line 342.
The field of dental prosthetics is very important because life expectancy has increased and a large group of patients use prosthetic restorations. In the case of prosthetic work, dental impressions pose a very serious threat as they are a breeding ground for bacteria. They often contain blood on their surface, which poses another risk of infection. Therefore, it would be necessary to supplement the publication with information related to this field. The authors should expand their work to include issues related to dental prosthetics. In addition to sterilizing tools, the process of taking an impression should be included. The issue of denture repair is also important, as dental technicians are at risk of infections related to the handling of dentures or other prosthetic restorations. Author Response : Thank you for your comments. Yes, that is a very important point. In Korea, infection control guidelines for the dental field had been developed, but their utilization rate was low. Therefore, the authors conducted the aforementioned study to incorporate the dental field into the national infection surveillance system, and hand hygiene surveillance activities will commence starting in 2024. Furthermore, as you pointed out, we are continuously working with national agencies to develop policies addressing the vulnerability in dental prosthesis management. As a result, the section on dental prosthesis management is being added to the supplementary guidelines for the Korean dental field. However, since this study was conducted in 2023 with the specific purpose of incorporating the dental field into the national infection surveillance system, it was not possible to include all the results related to the point you raised in this paper. Therefore, references 20, 21, and 22 were added at the beginning of the discussion section.
Comments 2: Additionally, the introduction should be more extensive and include more literature. Currently, it only has 7 items of literature. Author Response : Thank you for your excellent feedback. First, We have strengthened the references in the introduction while staying within the overall scope of the content. [referece 4-13]
|
||
- Additional Notes
We sincerely hope that the revisions and the additional information we have provided sufficiently address the reviewer’s concerns. We are grateful for the opportunity to improve our manuscript.
Round 2
Reviewer 2 Report
Comments and Suggestions for Authors
Thank you very much for your responses and the changes you made. I recommend this work for publication.